# Tick-wildlife host-pathogen network interactions in Northern Africa

**Marta Rafael** [1]ʘ*, **Amalia Segura** [2]ʘ, **Rita Vaz-Rodrigues** [1], **David Relimpio** [1],
**Oscar Rodríguez** [2], **Gabriela de la Fuente** [3], **Julio Isla** [3], **Christian Gortázar** [1],
**José de la Fuente** [1,4]*

**1** SaBio. Instituto de Investigación en Recursos Cinegéticos IREC-CSIC-UCLM-JCCM, Ronda de Toledo 12, Ciudad Real, Spain, **2** BP 30, Sidi Allal el Bahraoui, Morocco, **3** Sabiotec, Edificio incubadora de empresas UCLM, Camino de Moledores s/n, Ciudad Real, Spain, **4** Department of Veterinary Pathobiology, Center for Veterinary Health Sciences, Oklahoma State University, Stillwater, Oklahoma, United States of America

ʘ These authors contributed equally to this work
* marta.srafael@uclm.es (MR); jose_delafuente@yahoo.com (JF)

## Abstract

Ticks are hosts and vectors of zoonotic pathogens, posing a critical threat to public health and the conservation of animal host populations, especially in Northern Africa. Tick-host-pathogen interactions are driven by tick spatial distribution and abundance, and the influence of biotic (animal hosts) and abiotic (environmental conditions) factors. The objectives of this study, conducted in the Maamora Forest (Northwest Morocco), were: (i) description of seasonal interactions network between off-host questing ticks and the wild hosts, rabbits (*Oryctolagus cuniculus*) and addax (*Addax nasomaculatus*), (ii) analysis of density-dependent and environmental effects in questing and on-rabbit ticks, and (iii) identification of tick-borne pathogens in questing and on-addax ticks. Results showed that questing and on-rabbit ticks (*Hyalomma lusitanicum, Rhipicephalus pusillus,* and *H. aegyptium*) presented significant spatial and seasonal differences. Questing ticks were highly abundant in summer, but infestation on rabbits was higher in spring. Spatially, areas with contact between rabbits and ungulates showed the highest tick infestations during summer. Ticks from rabbits were density-dependent and had a positive relationship with questing ticks. Addax was infested by *H. lusitanicum* ticks. Tick network of interactions resulted in the presence of *Coxiella burnetii* in both questing and in addax ticks (17–27%), and *Rickettsia aeschlimannii* in *H. lusitanicum* questing ticks (4%). These results support that ticks represent a challenge for human and animal health, as well as ecosystems in Northern Africa, emphasizing the need for long-term studies on their network of interactions, seasonal activity patterns, and tick-borne pathogens in wildlife.

**Data availability statement:** All relevant data are within the manuscript and its Supporting information files.

**Funding:** This research was partly supported by the 2022-GRIN-34227 grant, funded by the University of Castilla-La Mancha (UCLM), Spain and EU-FEDER. R. Vaz-Rodrigues was supported by a doctoral contract (2022-PRED-20675), from UCLM and co-financed by the European Social Fund (ESF).

**Competing interests:** The authors have declared that no competing interests exist.

## Introduction

Ticks are important ectoparasites as vectors and reservoirs of pathogenic bacterial, viral, and protozoan microorganisms [1]. As a result, intricate eco-epidemiological webs for pathogen transmission can emerge, potentially impacting both animal and human health. Unlike other ectoparasites, ticks present a distinctive life cycle involving one or multiple hosts throughout their lifespan. Depending on whether the tick life cycle is of one, two, or three hosts, they pass through various host species. Immature stages, such as larvae and nymphs, tend to parasitise animals of a smaller size, like lagomorphs, rodents, or lizards. In contrast, adult stages parasitise large to medium-sized animals, for instance, camels and ungulates [2]. In between life stages, ticks molt into the next stage either on the first host species (one-host cycle) or in the environment (two or three-host cycle). Hence, between each life cycle, off-host ticks can be identified questing, a host-seeking behaviour where hard ticks ascend vegetation and wait for a passing host to attach [3]. Ticks' abundance and spatial distribution are highly dependent on biotic factors, including host population, individual host characteristics and the abundance of potential hosts, typically gathering in environments with higher host aggregation [4,5], as well as abiotic environmental variables such as temperature and seasonality [6,7]. In North Africa, ticks share a Mediterranean climate with four seasons, characterised by hot and dry summers and wet winters, supporting a rich diversity of tick species and tick-borne pathogens [8]. Numerous wildlife species have been recorded as hosts, potentially transporting ticks and pathogens into and out of North Africa through migratory birds or traded animals (livestock and exotic pets). For instance, *Hyalomma (H.) aegyptium* ticks have been recorded on *Testudo graeca* spur-thighed tortoises (considered as an exotic pet) in Morocco, Algeria and Turkey [9–11]. In mammals, *H. dromedarii*, *H. marginatum* or *H. dentritum* have been found on addax in Morocco and Tunisia [12–14]. Moreover, *H. excavatum* and *H. dromedarii* adult ticks have been reported on scimitar-horned oryx in Tunisia [13]. Nevertheless, little is known about the role of certain wild species as reservoirs, such as small mammals like the wild rabbit *Oryctolagus cuniculus*. These have been documented in Egypt and other Mediterranean environments as a carrier of *Coxiella burnetii* associated with *H. lusitanicum* [15,16]. Furthermore, wild rabbits, as highly abundant populations and prolific breeders, may pose a higher risk of transmitting the infection [17]. As the number of susceptible hosts for the same tick species increases, the transmission of numerous infectious agents across different species is facilitated, establishing intricate networks for pathogen transmission, leading to significant epidemiological implications [18]. Indeed, ticks from North Africa have been associated with several infectious agents, such as *Anaplasma phagocytophylum*, *Babesia bovis*, *B. caballi* and *B. divergens*, *Borrelia lusitaniae* and *B. burgdorferi*, the Crimean-Congo haemorrhagic fever virus (CCHFV), *C. burnetii*, *Ehrlichia ewingii*, *Rickettsia aeschlimannii*, *R. africae* and *R. massiliae*, *Theileria annulate*, among others [2,11,19,20]. Furthermore, in North African countries, ticks have also been previously reported in humans, such as *Ixodes ricinus*, *H. aegyptium, H. marginatum*, *Rhipicephalus bursa* or *R. sanguineus*, and present a zoonotic potential for anaplasmosis, borreliosis, ehrlichiosis, Q fever, rickettsiosis, babesiosis or CCHFV [5,20–22].

In this study, we investigate the network interactions between off-host ticks (present in vegetation searching for a host, hereafter questing ticks), wild rabbits as intermediate hosts and ungulates in Maamora forest (Northwest Morocco), where livestock is abundant [23]. The Saharan antelope addax (*Addax nasomaculatus*), a "critically endangered" ungulate [24], has been identified as a carrier for several tick-borne pathogens, including *Rickettsia* spp., *Ehrlichia* spp., *Anaplasma* spp. and *C. burnetti* [14]. Hence, this species was selected as the ungulate animal model to study tick life cycle transmission, aiming to increase the knowledge of potential tick-borne pathogens affecting this species (e.g., plausibly *C. burnetii,* causing abortion). Particularly, this study intends to i) determine the abundance and diversity of questing ticks by flagging and ticks collected from dead rabbits seasonally, ii) determine density-dependent and environmental effects in questing ticks and ticks from wild rabbits, and iii) determine tick-borne pathogens circulating in questing ticks and those infecting addax. Tick-borne pathogens were detected by conventional PCR techniques, aimed at *Anaplasma* spp., *Babesia* spp., *C. burnetii*, *Rickettsia* spp., *Theileria* spp. and CCHFV, previously described pathogens in Africa [2,19,20] and in the study area [11,14], followed by a phylogenetic analysis of positive results. This study results provide information to advance in a One-Health approach aimed at preventing the transmission of zoonotic diseases across trans-Mediterranean regions impacting both Europe and Northern Africa [20].

## Materials and methods

### Study site

The study took place on a private protected reserve (3000 ha) within the Maamora cork oak Forest (Northwest Morocco), characterised by a varied and rich undergrowth (i.e., high species richness and cover), evidenced by its variety of species (Mediterranean brooms [*Genista linifolia, Cytisus arboreus*, *Stauracanthus genistoides*], dwarf palm [*Chamaerops humilis*], French lavender [*Lavandula stoechas*], sparrow-wort [*Tymelea lytroides*] and sage-leave rockrose [*Cistus salviifolius*]) and surface coverage, compared to other unprotected areas in Maamora which are highly overgrazed by livestock [25]. The undergrowth and mammal density varies in the reserve. Therefore, our study was conducted in four zones, each 5 km apart (A, B, C and D; Fig 1). All zones have similar scrub diversity but differ in scrub cover: over 50% in B and C, over 40% in A and over 25% in D. Bare ground was absent in zones B and C, while it exceeded 15% in zones A and D. Although there are rabbits in the four study zones, ungulates are absent in zones B and D due to fencing that separates these areas. The density of addax in A and B zones is 25 and 44 individuals (in 800 ha each), respectively. Apart from addax, there are other tick hosts present in the study area, such as spur-thighed tortoises, which have been documented

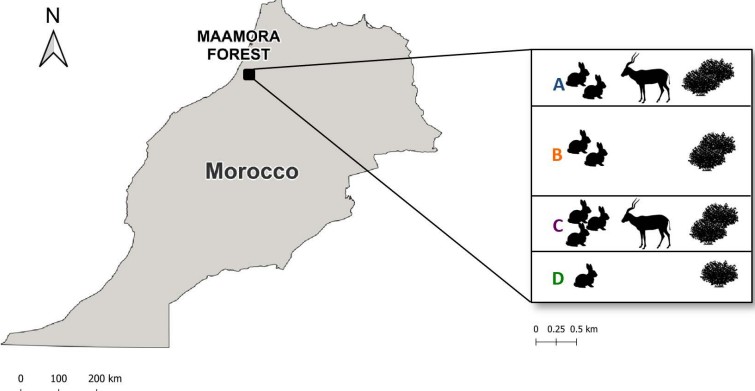

**Fig 1. Study area.** Representation of the study area, divided into zones A, B, C, and D, and abundance of vegetation (scrub), rabbits and addax across the study area. Rabbits' abundance and scrub cover vary throughout the study area, and there are no ungulates in zones B and D. The map was generated using the ArcGIS software, version 10.8.

to present high tick infestation intensity [mean of 4 *H. aegyptium* ticks/tortoise; [9,11]], as well as other plausible hosts like hedgehogs. As a private initiative for the addax conservation program, a 33–62% serological antibody prevalence of *C. burnetii* has been registered in other wild ungulates inhabiting the study area.

**Tick sampling and rabbit density**

Questing ticks were collected in the four study zones in June 2022, March, June, and November 2023, corresponding to spring, summer and autumn seasons. Three individuals conducted the flagging method by dragging the cloth across ground-level vegetation, with a sampling effort of one hour per study zone [2].

Ticks infesting rabbits were collected from five randomly selected dead individuals in each sampling zone in the same periods as questing ticks. Dead rabbits were the result of traffic crashes and naturally dead animals. In addition, coinciding with study periods and zones, rabbit populations were surveyed at night, using spotlights while driving at speeds of 20–30 km/h through human tracks along 4 transects of 3–5 km length and 30 m wide on non-brightest nights (full moon). The collected data were georeferenced to determine rabbit densities and spatial distribution. Hotspot analysis for determining aggregation patterns was accomplished by GIS (ArcGIS 10.8).

Ticks infesting addax, as a potential ungulate model for tick infestation, were collected from ten live animals during the annual sanitary control in November 2022. The aim was to determine their role in the network interaction of ticks and to explore vegetation and host-associated tick-borne pathogens.

Ticks collected from vegetation, wild rabbits and addax were placed into plastic vials containing RNAlater (Thermo Fisher, Waltham, MA, USA) and stored at −20 ºC until further analysis.

Ticks' collection from vegetation, as well as from dead and live animals, was carried out as part of a private initiative for monitoring threatened species in the Maamora Forest, following approved ethical protocols for wildlife capture and management (no permits were required).

**DNA/RNA isolation and PCR for pathogen detection in questing and addax ticks**

A representative subsample of collected ticks was used for DNA extraction and analysis of infectious agents (S1 Table). Ticks were morphologically identified using available taxonomic keys [2]. Nucleic acid extraction was accomplished from individual and tick pools (mean of 3 ticks/pool). Samples were pooled randomly, ranging from 1 to 6 ticks/pool. DNA and RNA were extracted from the internal tissues of ticks, discarding the external cuticle, using 250 µL of TRI Reagent (Sigma-Aldrich, St. Louis, USA), and following the manufacturer's instructions. The concentration (ng/µL) and purity of samples were evaluated using a Nanodrop One spectrophotometer (Thermo Scientific, Waltham, USA), through the quantification of the nucleic acids at an optical density of 260 nm ($OD_{260}$), and the ratio of absorbance at 260/280 nm. The quality of the extraction protocol and morphological identification of tick species was confirmed by DNA barcoding of the *mitochondrial 16S ribosomal DNA* (*16S rDNA*) gene and the *cytochrome oxidase subunit I* (*COI*) gene (Table 1). Ticks used for molecular identification included four individual adult ticks collected from vegetation, confirming also the identification of addax's ticks, and ticks collected from rabbits – four individual adult ticks, and three pools of nymphs, constituted by two to four nymphs.

A random sample of addax's ticks (n = 49 ticks, 11 tick pools) and ticks from vegetation (n = 84 ticks, 24 pooled ticks) were tested using conventional polymerase chain reaction (PCR) aimed at detecting the presence of pathogenic agents, such as *Anaplasma* spp., *Babesia* spp., *C. burnetii*, *Rickettsia* spp., *Theileria* spp. and a nested reverse transcription (RT)-PCR for the identification of the CCHFV. Table 1 provides information on the specific targeted regions for each PCR assay, the protocol, and the specific primers used.

The PCR reactions were performed in a 25 µL volume, including 12.5 µL of PCR Master Mix 2x (Promega, Madison, WI, USA), 1 µL of each primer (10 µM working solution), 9 µL of RNase-free water (Thermo Scientific), and 1.5 µL of DNA sample. For the nested RT-PCR assessment of CCHFV, the commercial kit Access RT-PCR System (Promega, Fitchburg, WI, USA) was used according to the manufacturer's instructions. The PCRs were conducted in a C1000 touch PCR

**Table 1. Primers and PCR conditions. List of primers and PCR protocol according to the pathogenic agent analysed.**

| Pathogen and/or target gene | Sequence 5'-3' (F: Forward/ R: Reverse) | Fragment (base pairs) | Annealing (ºC) | Reference |
|---|---|---|---|---|
| *16S rDNA* | F: CCGGTCTGAACTCAGATCAAGT<br>R: CTGCTCAATGATTTTTTAAATTGCTGTGG | 460 bp | 48 | [14] |
| *COI* | F: GGTCAACAAATCATAAAGATATTGG<br>R: TAAACTTCAGGGTGACCAAAAATCA | 650 bp | 50 | [26] |
| *Anaplasma* spp.<br>(*16S rRNA*) | F: CAGAGTTTGATCCTGGCTCAGAACG<br>R: GAGTTTGCCGGGACTTCTTCTGTA | 421 bp | 42 | [27] |
| *Anaplasma* spp. (*msp5*) | F: GCATAGCCTCCGCGTCTTTC<br>R: TCCTCGCCTTGGCCCTCAGA | 456 bp | 54 | [27] |
| *Anaplasma* spp. (*msp4*) | F: CGGATCCTTAGCTGAACAGGAATCTTGC<br>R: GGGAGCTCCTATGAATTACAGAGAATTG<br>TTTAC | 849 bp | 60 | [27] |
| *Coxiella burnetii*<br>(*IS111a*) | F: CAAGAATGATCGTAACGATGCGC<br>R: CTCGTAACACCAATCGCTTCG | 349 | 63 | [28] |
| Crimean-Congo Haemorrhagic Fever vírus<br>(S segment) | F1: TTGTGTTCCAGATGGCCAGC<br>R1: CTTAAGGCTGCCGTGTTTGC<br>F2: GAAGCAACCAARTTCTGTGC<br>R2: AAACCTATGTCCTTCCTCC | 211 | 60 and 57 | [29] |
| *Rickettsia* spp.<br>(*16S rRNA*) | F: AGAGTTTGATCCTGGCTCAG<br>R: AACGTCATTATCTTCCTTGC | 416 | 54 | [14] |
| *Rickettsia* spp. (*ompA*) | F: ATGGCGAATATTTCTCCAAAA<br>R: AGTGCAGCATTCGCTCCCCCT | 630 | 54 | [30] |
| *Rickettsia* spp. (*ompB*) | F: GGGTGCTGCTACACAGCAGAA<br>R: CCGTCACCGATATTAATTGCC | 618 | 53 | [31] |
| *Rickettsia* spp. (*gltA*) | F: GGGGGCCTGCTCACGGCGG<br>R: ATTGCAAAAAGTACAGTGAACA | 360 | 45 | [32] |
| *Rickettsia* spp. (*recA*) | F: TGCTTTTATTGATGCCGAGC<br>R: CTTTAATGGAGCCGATTCTTC | 428 | 52 | [31] |
| *Rickettsia* spp. (*atpA*) | F: ACATATCGAGATGAAGGCTCC<br>R: CCGAAATACCGACATTAACG | 731 | 48 | [31] |
| *Babesia/ Theileria* spp. (18S rRNA) | F: GTCTTGTAATTGGAATGATGG<br>R: CCAAAGACTTTGATTTCTCTC | 550 | 55 | [33] |

thermal cycler (Bio-Rad, Hercules, CA, USA), with the specific PCR fragments visualised in a 1.5% agarose gel stained with GelRed (Biotium, Fremont, CA, USA) under UV transillumination.

## Phylogenetic analysis

Suspected PCR-positive samples were purified and sequenced using the Sanger method at Secugen (Madrid, Spain). Sequences were assessed in Chromas software (version 2.6.6.), and homology analysis was conducted using the National Center for Biotechnology Information (NCBI) database, employing the Basic Local Alignment Search Tool (BLAST). The newly obtained sequences were deposited in the GenBank database. Multiple sequence alignment was carried out between the obtained sequences and reference sequences from GenBank using the Multiple Sequence Comparison by Log-Expectation (MUSCLE) algorithm. Phylogenetic analysis was performed in MEGA software (version 11.0.13) to evaluate the genetic association between retrieved sequences and reference samples. The best-fit model was selected based on the corrected Akaike Information Criterion (cAIC), and phylogenetic trees for positive infectious agents were generated using Maximum Likelihood and Neighbor-Joining methods. To ensure the reliability of the produced trees, 1000 bootstrap replicates were implemented.

## Statistical analysis

The prevalence of questing ticks and rabbit ticks was calculated to analyse differences among seasons and study zones by Pearson's Chi-squared test ($\chi^2$) and Fisher's exact test, respectively, both followed by a post-hoc test (Bonferroni adjustment). The relationship between the response variable rabbit ticks and predictors rabbit population density, season, spatial distribution (zones) and questing ticks was performed using a generalized linear model (GLM) with a Poisson distribution and logarithmic link function. The most parsimonious model was then selected using a forward stepwise procedure based on Akaike Information Criteria (AIC) [34]. Statistical analyses were performed with R software, version 4.2.3 [35].

## Results

### Questing ticks, rabbit ticks and rabbit densities

Questing ticks (n = 3754) were mostly adults (1966 females and 1787 males) and one nymph. They were morphologically identified as *H. lusitanicum* (n = 3653), *H. aegyptium* (n = 98), and *Rhipicephalus* spp. (n = 3).

Seasonally, higher numbers of questing ticks were collected in summer 2023 and 2022 (1520 and 1230, respectively), when compared with spring (n = 883) and autumn (n = 121) (Fig 2A). Spatially, more questing ticks were collected in zones C and A (1948 and 1127, respectively) than in B and D (631 and 48, respectively).

Our results indicate significant differences in the number of questing ticks collected throughout the different study zones and seasons ($\chi^2$ = 224.68, *p*-value <0.05; Fig 2B). Fig 2B illustrates the percentage of ticks collected across sampling zones and seasons and highlights the significant relations between collected and expected ticks in spring 2023 in zones A and C, summer 2023 in zone B, and autumn 2023 in zone D.

Rabbit ticks (n = 1137) were mainly adults (387 males and 200 females), nymphs (n = 474) and larvae (n = 76). Mean infestation rate of 18 ± 9.43 ticks/rabbit. Ticks were morphologically classified as *H. lusitanicum* (4 adults), *Hyalomma* spp. (347 nymphs and larvae) and *Rhipicephalus pusillus* (583 adults, 185 nymphs), and *Rhipicephalus* spp. (18 larvae).

Seasonally, more ticks were collected in spring 2023 (n = 417) and summer 2023 and 2022 (n = 391 and 272, respectively), compared with autumn (n = 57) (Fig 3A). Spatially, a larger number of ticks were collected in zone B (n = 363), compared with zones A, D and C (n = 275, 262 and 237, respectively).

Our results show significant differences in the number of ticks collected from rabbits throughout the different study zones and seasons ($\chi^2$ = 388.32, *p*-value < 0.05, Fig 3B). Rabbit ticks were significantly higher in summer 2022 for zones B and C, in spring 2023 for zone D, in summer 2023 for zone A, and in autumn 2023 for zone D (Fig 3B).

Rabbit densities varied from 7 to 26 individuals/km² in the four zones and seasons, with zone D presenting slightly lower densities compared with the other zones. Spring, followed by autumn, shows the highest densities (S1 Fig). Rabbits were aggregated in different zones according to season and year. In both summers, the aggregation was observed in zones C and D, in spring in zones A, C and D and in autumn in zones A and B (S2 Fig).

Model analysis revealed that tick parasitisation in rabbits was significantly related to rabbit densities, season, zone and questing ticks (Table 2). Higher density of rabbits and questing ticks increases rabbit parasitisation.

### Tick molecular identification and phylogeny

The morphological identification of ticks was confirmed by barcoding analysis of individual ticks collected from vegetation (questing) and attached to wild rabbits, corroborating their previous classification. Questing ticks (Isolate numbers 8, 12 and 15) were molecularly identified as *H. lusitanicum*. BLAST analysis revealed a sequence sharing of 99.78–99.80% nucleotide identity with the *16S rDNA* and/or *COI* genes of *H. lusitanicum* ticks attached to rabbits (GenBank accession number MZ420716) or unattached in vegetation (GenBank accession number EU827708). Adult ticks collected from rabbits (Isolate numbers 34, 38, 39 and 40) were molecularly identified as *R. pusillus* with a 99.47–100% nucleotide identity to the *16S rDNA* gene (GenBank accession numbers Z97883 and OL454849). On the other hand, nymphs (Isolate

(A)

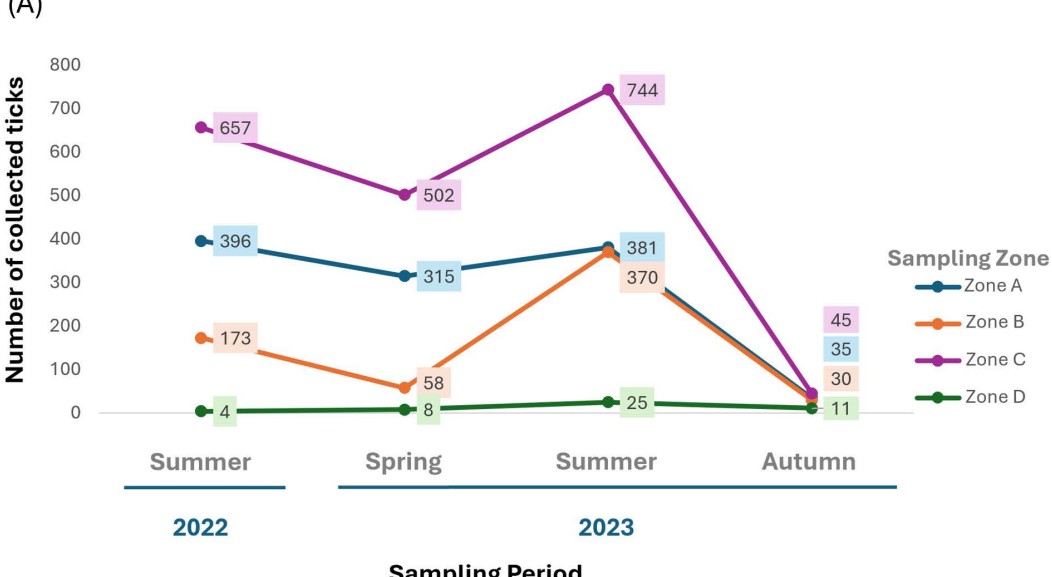

(B)

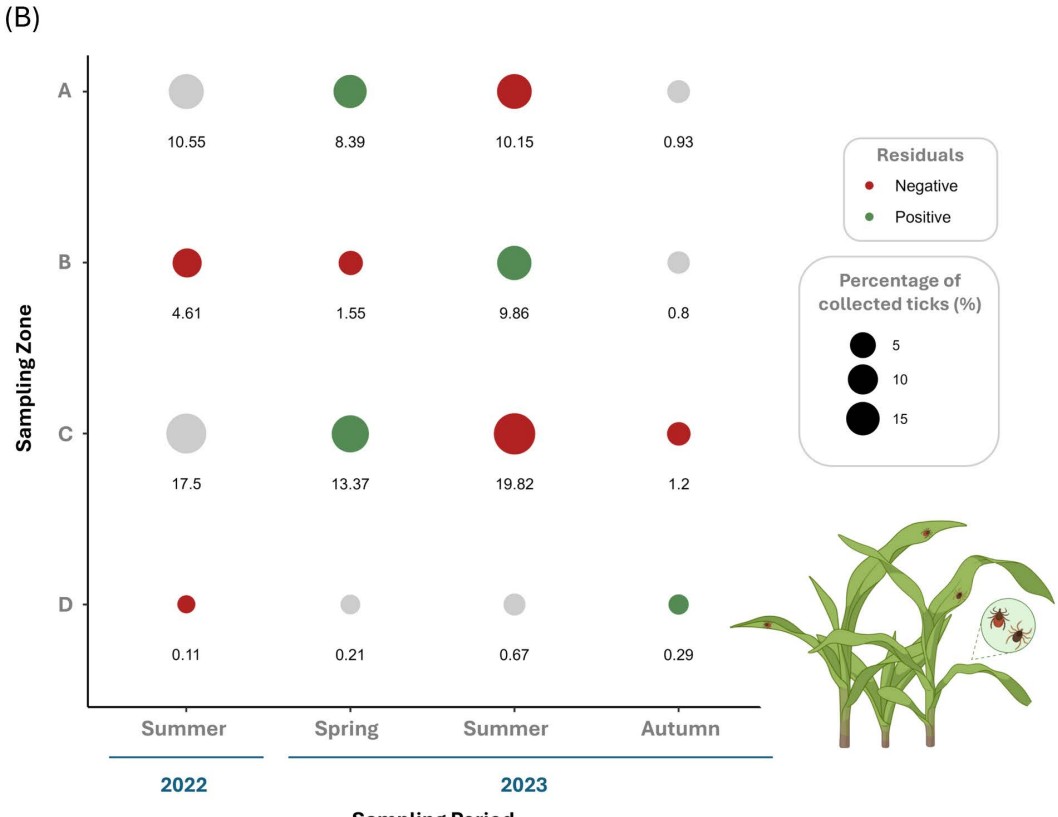

**Fig 2. Questing ticks.** (A) Seasonal and spatial distribution of collected questing ticks across zones A, B, C, and D and seasons. (B) Percentage of questing ticks (size of dots), representation of residuals and adjusted *p*-values from Pearson's Chi-Squared Test post-hoc analysis. Residuals indicate higher (green) or lower (red) values than expected, corresponding to statistically significant correlations (*p*-values < 0.05). Non-statistical results are shown with a grey dot.

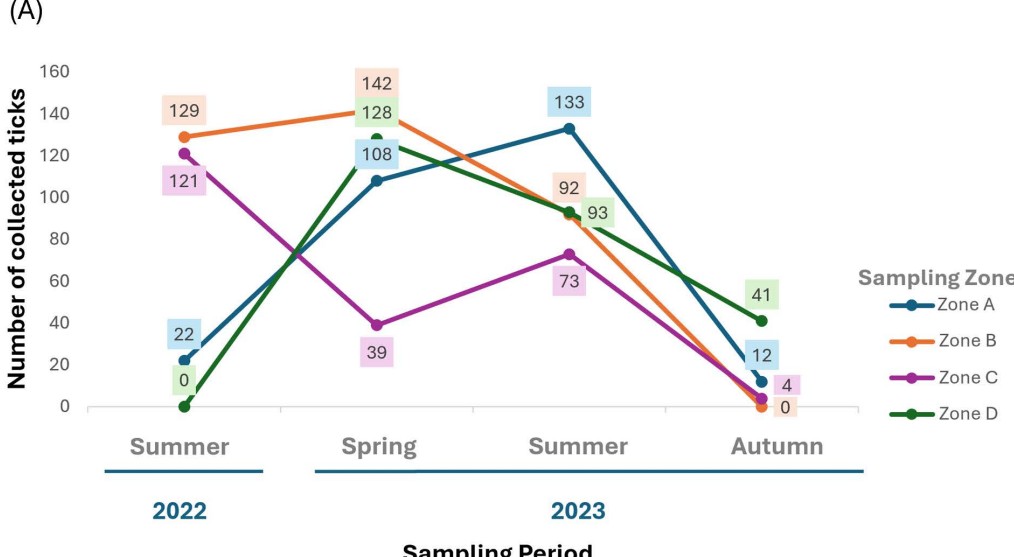

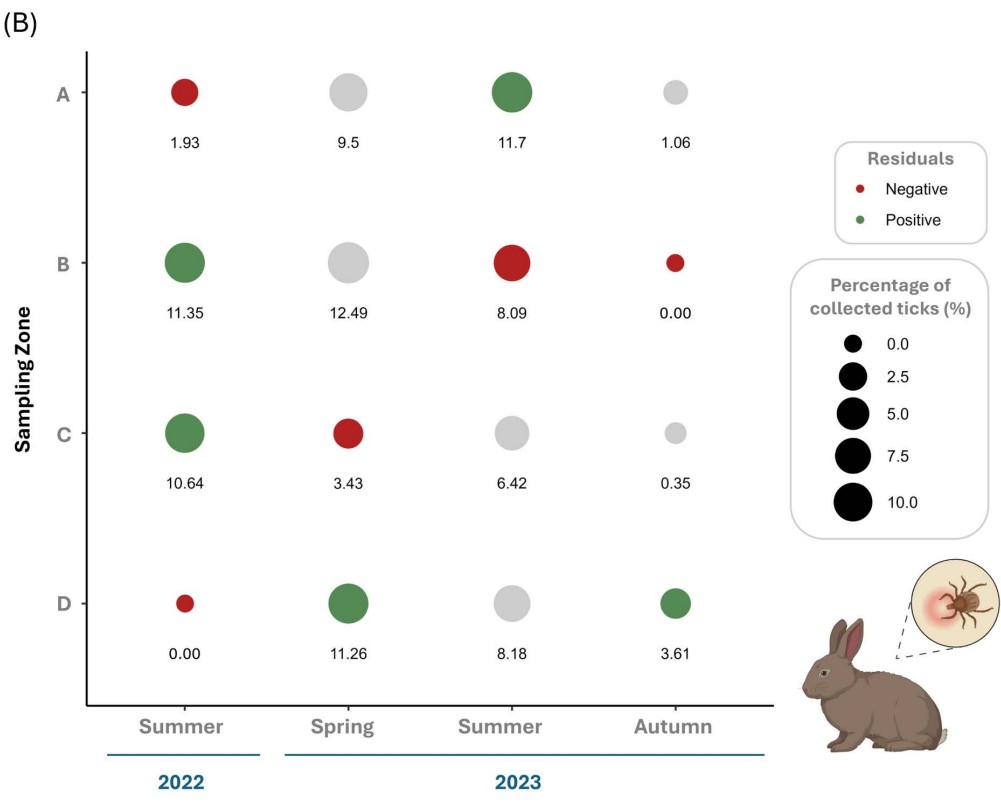

**Fig 3. Wild rabbit (*Oryctolagus cuniculus*) (A) Seasonal and spatial distribution of collected rabbits' ticks across zones A, B, C, and D and seasons. (B)** Percentage of ticks collected from wild rabbits (size of dots), representation of residuals and adjusted p-values from Fisher's exact test post-hoc analysis. Residuals indicate higher (green) or lower (red) values than expected, corresponding to statistically significant correlations (*p*-values < 0.05). Non-statistical results are shown with a grey dot.

**Table 2. Statistical parameters of the generalised linear model (GLM) used to explain rabbit ticks according to rabbit density, questing ticks, season and spatial distribution (zone).** See S2 Table for model selection.

| | Estimate | Std error | Z value | p-value |
|---|---|---|---|---|
| **Rabbit density** | 0.077 | 0.009 | 8.854 | <0.01 |
| **Questing ticks** | 0.002 | 0.001 | 3.093 | <0.01 |
| **Spring23** | 1.505 | 0.154 | 9.741 | <0.01 |
| **Summer22** | 0.855 | 0.179 | 4.756 | <0.01 |
| **Summer23** | 1.579 | 0.188 | 8.374 | <0.01 |
| **Zone B** | 0.501 | 0.104 | 4.787 | <0.01 |
| **Zone C** | −0.555 | 0.147 | −3.764 | <0.01 |
| **Zone D** | 0.749 | 0.178 | 4.196 | <0.01 |

numbers 29 and 42) shared a 99.34–100% nucleotide identity with the *H. lusitanicum 16S rDNA* and/or *COI* genes (GenBank accession numbers OR091373 and MK946446). Phylogenetic analysis was assessed through the correlation between sequenced samples, retrieved from vegetation and rabbit, and corresponding tick species *Hyalomma* and *Rhipicephalus* obtained from the GenBank database (Fig 4 and 5).

## Ticks and tick-borne pathogens in addax and questing ticks

A total of 116 adult ticks were collected from captured addax. Ticks were morphologically identified as *H. lusitanicum* (76 males and 39 females) and *H. marginatum* (n = 1 male). One individual had *H. lusitanicum* and *H. marginatum*.

Tick-borne pathogens were detected in 20.8% (5/24) of the analysed pooled questing ticks and 27.3% (3/11) of the pooled addax ticks. The pathogen DNA detected in questing ticks corresponded to *Rickettsia* (4.1%) and *Coxiella* (16.7%) species and, in ticks retrieved from addax, *C. burnetii* (27.3%). The *16S rDNA* PCR-positive sample (Isolate 23) to *Rickettsia* spp. shared 100% nucleotide identity with *R. aeschlimannii* retrieved in Spain from *H. rufipes* and *H. marginatum* ticks (GenBank accession numbers MW394393 and MW394392, respectively) (Fig 6). This corresponds to the first record of *R. aeschlimannii* in *H. lusitanicum* free-roaming ticks in vegetation. In addition, BLAST analysis of four tick pools from vegetation (Isolates 13, 14, 16 and 22) and three from addax (Isolates 2A, 6A and 10A) revealed a 99.12–100% nucleotide identity to the *IS1111* gene of *C. burnetii* (GenBank accession numbers MG764422, MZ673295 and CP014565). The genetic correlation between isolate samples and *C. burnetii* sequences retrieved from the GenBank database was assessed through phylogenetic analysis (Fig 7). Concerning pathogen detection, questing ticks and addax tick samples were PCR-negative for *Anaplasma* spp., *Babesia* spp., *Theileria* spp. and CCHFV infectious agents.

## Discussion

Our results describe the annual distribution of multiple tick species – *H. lusitanicum*, *R. pusillus*, and *H. aegyptium* – found in vegetation and wild hosts, resulting in significant spatial and seasonal differences. While questing ticks reached their highest abundance in summer, ticks from wild rabbits were most abundant in spring. Ticks from rabbits were density-dependent and showed a positive relationship with questing ticks. Tick network interactions led to the presence of infectious agents present in both questing and addax ticks, with *C. burnetii* found in both, and *R. aeschlimannii* in questing ticks (Fig 8).

The number of ticks collected in vegetation and hosts varies considerably between seasons and years due to abiotic factors [6,7]. Several tick genera decreased their abundance in winter months and increased it in summer [14]. Our study supports this variation in questing ticks, with the predominant *H. lusitanicum* showing a difference of 1400 individuals when comparing summer with autumn. Accordingly, [36] have documented higher activity from May to July in *H. lusitanicum* adult ticks, which might increase infection rates of hosts in summer in North Africa. In our study, *R. pusillus* was the

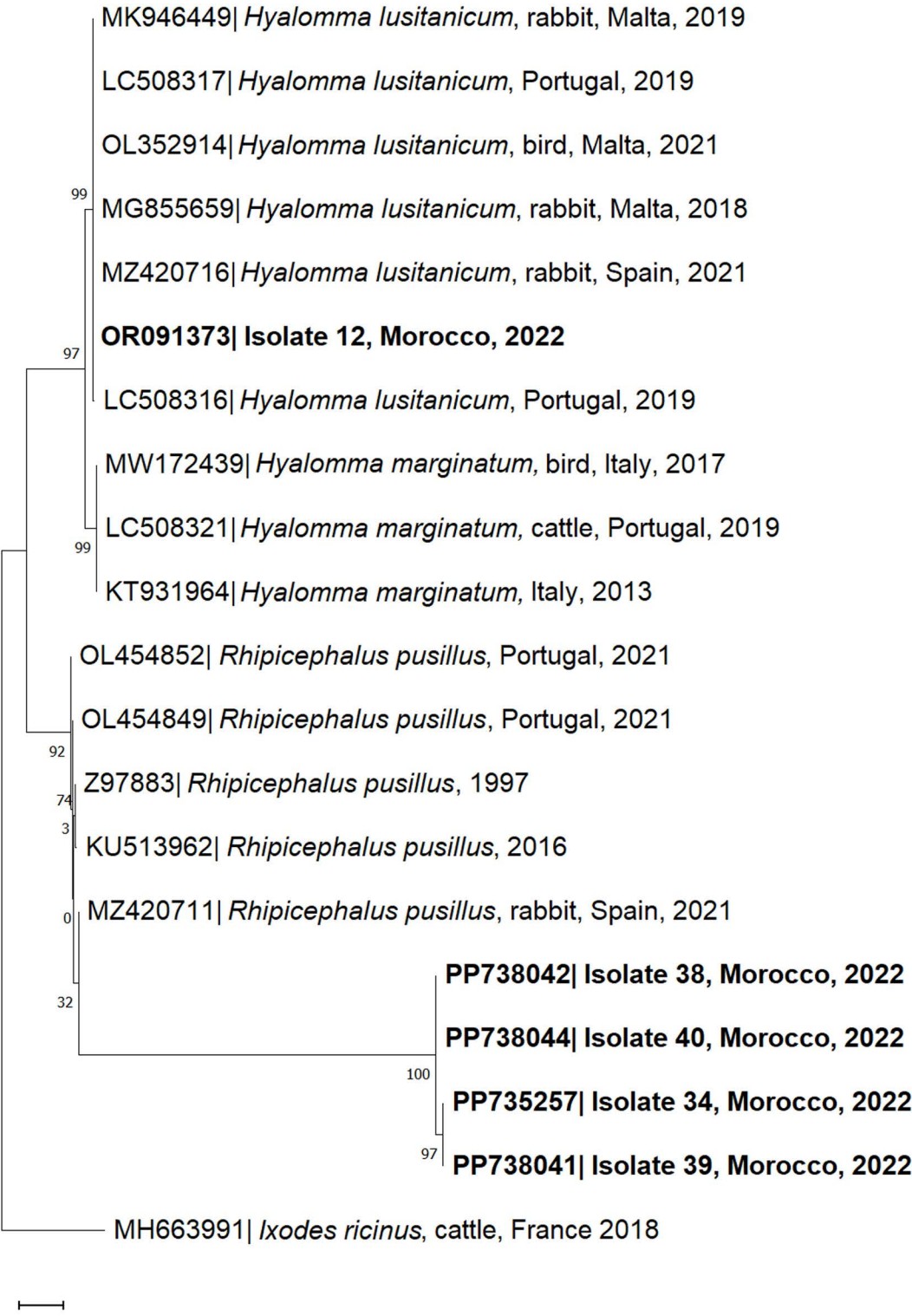

**Fig 4. Phylogenetic tree of the *16S rDNA* sequences of *Hyalomma lusitanicum* isolated from ticks collected from vegetation (isolate 12) and *Rhipicephalus pusillus* isolated from ticks collected from wild rabbits (*Oryctolagus cuniculus*) (isolates 34, 38, 39, 40).** The analysis

was obtained based on the Neighbor-Joining method with a Tamura-3-parameter model with a discrete Gamma distribution. The characterised species in this study are represented in bold. Sequence names include the GenBank accession number, organism name, host (if mentioned), country of origin and year of collection or submission. The reliability of internal branches was assessed using the bootstrapping method with 1000 replicates.

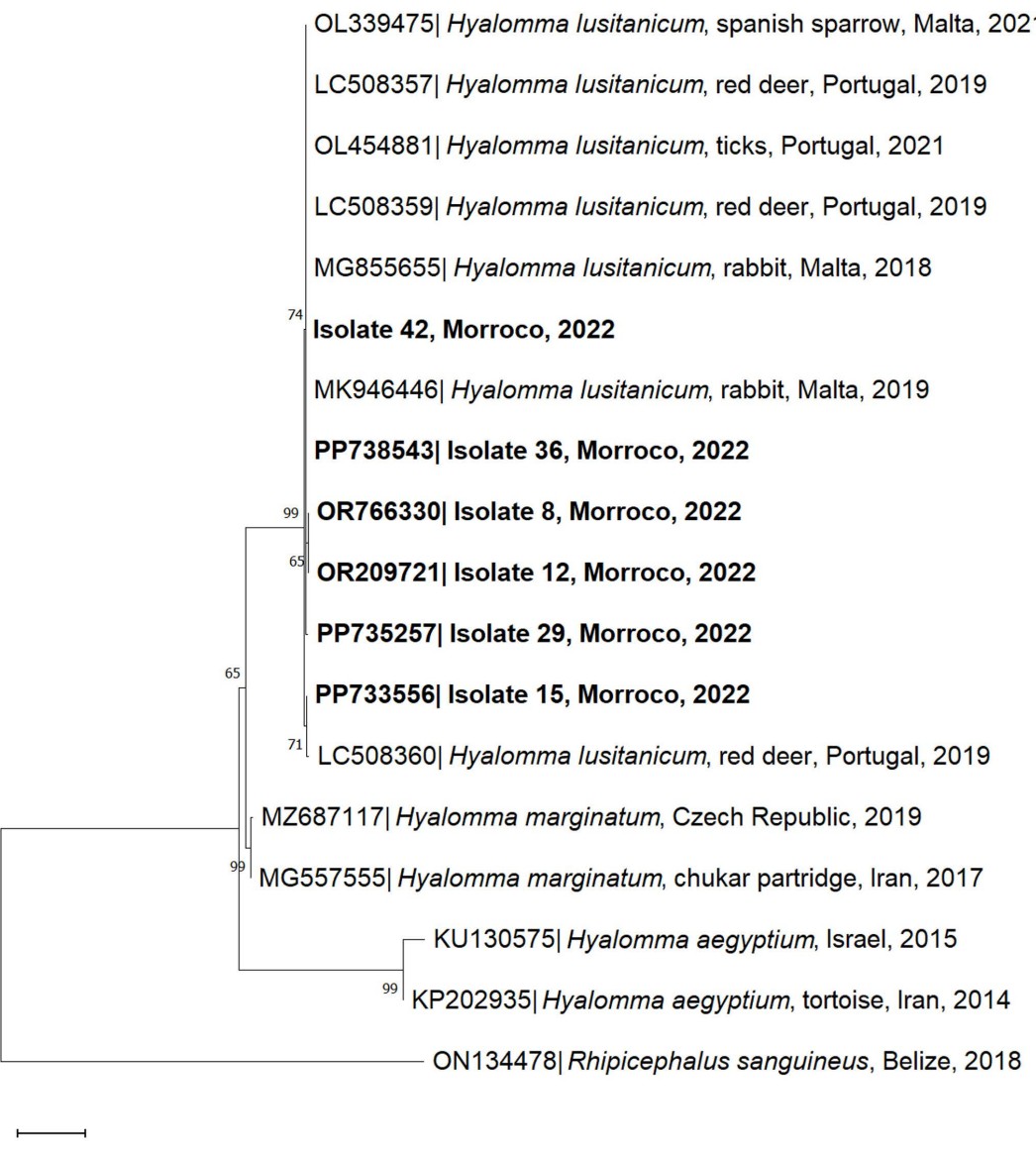

**Fig 5. Phylogenetic tree of the *cytochrome oxidase subunit I* (*COI*) sequences of *Hyalomma lusitanicum* isolated from questing ticks (isolates 8, 12 and 15) and from wild rabbits (*Oryctolagus cuniculus*) (isolates 29 and 42).** The analysis was obtained based on the Neighbor-Joining method with Tamura 3 parameters with invariable sites. The characterised species in this study are represented in bold. Sequence names include the GenBank accession number, organism name, host (if mentioned), country of origin and year of collection or submission. The reliability of internal branches was assessed using the bootstrapping method with 1000 replicates.

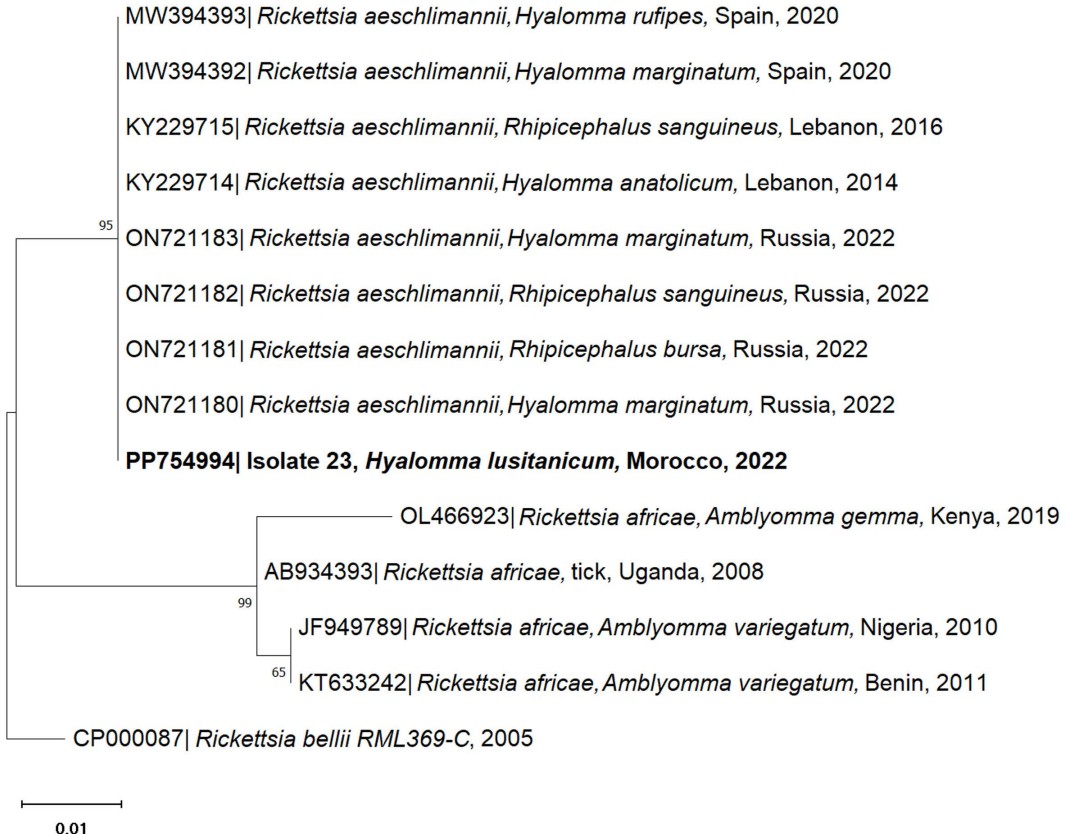

**Fig 6. Phylogenetic tree of the *16S rDNA* sequence of *Rickettsia aeschlimannii* isolated from *Hyalomma lusitanicum* questing ticks (isolate 23).** The analysis was obtained based on the Neighbor-Joining method with a Jukes-Cantor model. The characterised species in this study are represented in bold. Sequence names include the GenBank accession number, organism name, host (if mentioned), country of origin and year of collection or submission. The reliability of internal branches was assessed using the bootstrapping method with 1000 replicates.

predominant species, followed by *H. lusitanicum,* in wild rabbits. In addition, the infestation was higher in spring (21 ticks/rabbit) than average (15 ticks/rabbit), as occurred in other Mediterranean areas [16]. *Rhipicephalus pusillus* life cycle is highly dependent on the presence of lagomorphs, increasing rabbit parasitation rates with rabbit densities-dependent effects [2,37]. *Hyalomma lusitanicum* network benefits from rabbit density effects, and its presence in addax closes the cycle between vegetation, wild rabbits and ungulates, as described by [5]. In Mediterranean areas, *H. lusitanicum* in rabbits has been documented to increase in zones where ungulates inhabit with wild rabbits [16]. Indeed, in our study, this occured in summer when rabbit densities decreased. Nevertheless, in areas without ungulate-rabbit interactions, tick parasitisation increases in spring and autumn, which may be related to the high-density-dependent effect of rabbits. Abiotic factors, such as water bodies, woodlands or scrub cover, as well as biotic ones, such as high diversity of host species, all contribute to the *H. lusitanicum* cycle [4,38]. On the other hand, as a questing tick, *H. aegyptium* may be influenced by other potential wild hosts [9,11]. For instance, the density and distribution of tortoises and hedgehogs may help explain the spatial differences observed in *H. aegyptium*, as scrub cover alone does not seem to account for them [11]. In addition to seasonal variations in tick abundance, annual differences in abiotic conditions, such as dry and humid years, can also affect tick abundance [39,40]. Previous studies have identified rainfall as a key factor, with increasing population abundance following periods of high cumulative rainfall [41,42]. In fact, significant differences in precipitation between the two studied years resulted in higher questing and on-rabbit ticks in the wetter year (19 and 30% higher, respectively). Current

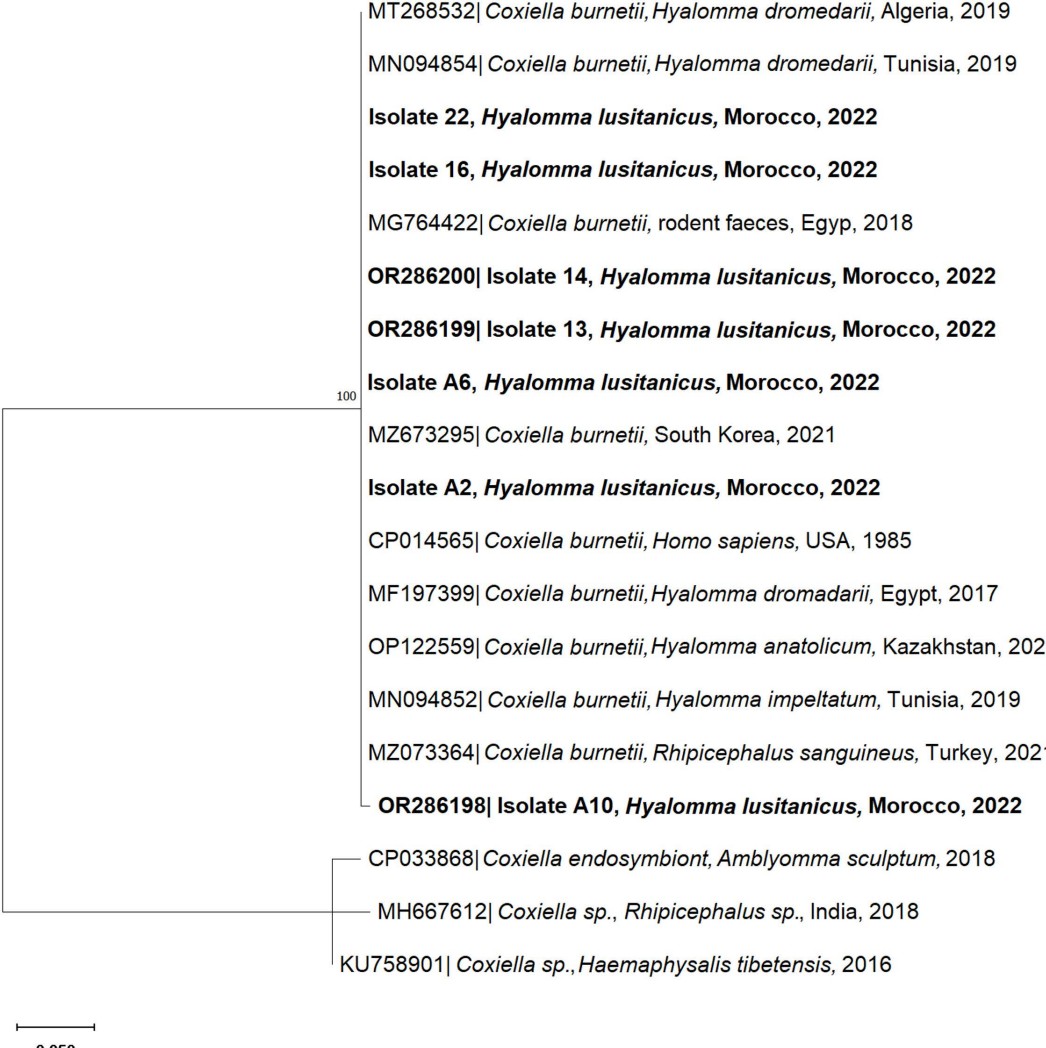

**Fig 7. Phylogenetic tree of the *IS1111* sequence gene of *Coxiella burnetii* isolated from *Hyalomma lusitanicum* questing ticks (isolates 13, 14, 16, 22) and attached to the Saharan antelope addax (*Addax nasomaculatus*; isolates A2, A6, and A10).** The analysis was obtained based on the Neighbor-Joining method with a Kamura 2 parameters model. The characterised species in this study are represented in bold. Sequence names include the GenBank accession number, organism name, host (if mentioned), country of origin and year of collection or submission. The reliability of internal branches was assessed using the bootstrapping method with 1000 replicates.

climate change models for the Mediterranean basin predict longer drought periods and more frequent, intense and unpredictable rainfall events [43–45]. Shifts in climate, trending towards warmer temperatures, may alter tick dynamics and seasonal activity patterns, resulting in increased activity in winter [39,46] and potentially the expansion of certain species, such as *H. marginatum* [39]. These climatic changes are likely to make larger regions of southern Europe more favourable for the expansion of North African ticks [47].

Acknowledging some limitations, such as tick-host dynamics and climatic conditions in a given year, we provide evidence of tick interactions between wild hosts and the importance of integrated One Health observational and modelling studies to detect changes in tick populations and disease transmission [20,40]. To our knowledge, we report the first detection of *R. aeschlimannii* in *H. lusitanicum* questing ticks in Africa, along with the identification of *C. burnetii*. *Rickettsia*

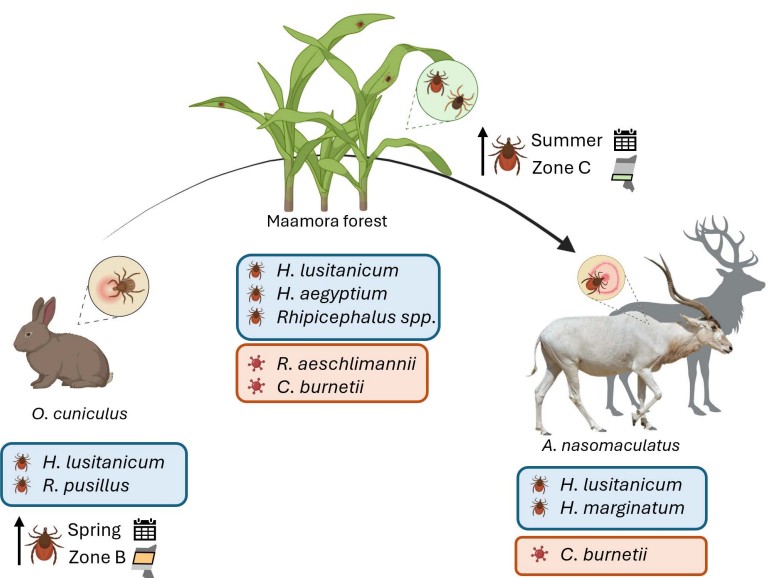

**Fig 8. Summary of this study major findings.** Representation of encountered ticks (Hyalomma lusitanicum, Hyalomma marginatum, and Rhipicephalus pusillus) and tick-borne pathogens (Rickettsia aeschlimannii, and Coxiella burnetii) in questing ticks and investigated hosts (wild rabbit and addax). In addition, the highest number of ticks collected from vegetation and rabbits according to sampling zone and period is also represented.

*aeschlimannii* belongs to the spotted fever group rickettsiosis, which includes potential zoonotic agents and is associated with the Mediterranean spotted fever-like disease in humans [48,49]. *Rickettsia aeschlimannii* is most commonly reported in *H. marginatum* ticks, even though it has also been detected in other tick species [12,20,50,51]. Studies conducted in other regions of North Africa point to a prevalence of tick-borne rickettsiosis between 0.5–64%, depending on the *Rickettsia* species [20]. In addition, in 2014–2017, a 50% prevalence of spotted fever group Rickettsia spp. was reported in *H. marginatum* ticks in addax in the same study area [14], with a higher prevalence (27.3%). Prevalence differences may be attributed to intraspecific tick characteristics, such as internal barriers (e.g., the tick immune system) or co-infection with undetected pathogens, which could influence ticks and their ability to sustain the infection [1].

*Coxiella burnetii* is the causative agent of Q fever in humans, ranging from mild symptoms, such as fever, to endocarditis in the chronic disease stage [52]. We detected a lower prevalence of *C. burnetii* (22.6% of questing ticks and 28.6% of addax ticks), compared to previous studies, which describe higher prevalences in the Mediterranean basin [16,53] or African countries [20]. Such differences may be attributed to different tick characteristics, the multitude of tick species and host communities present in both regions, which could condition the maintenance and/or increase the prevalence of the pathogen [1]. *Coxiella burnetii* in wildlife and domestic animals has been described in multiple tick species such as *Ixodes* spp., *Rhipicephalus* spp., *Amblyomma* spp., *Dermacentor* spp. or *Hyalomma* spp. [54]. In Africa, Q fever has a seroprevalence of 8% in humans; higher in agricultural countries (e.g., Algeria or Egypt with 15% and 16% seroprevalence, respectively) [54,55]. In cattle and small ruminants from Africa, antibody seroprevalence against *C. burnetii* ranges from 18% to 55% and 11% to 33%, respectively [20,55]. In addition to its importance in human health, *C. burnetii* infection in sheep can lead to abortion and cause reproductive losses in wild ungulates, such as red deer [56], suggesting that addax might also be affected. The absence of *Anaplasma* spp., *Babesia* spp., *Theileria* spp. and CCHFV in *H. lusitanicum* stands in contrast to the prevalence of these pathogens in *H. marginatum* found within the addax population of the studied area [14]. These differences may be attributed to factors such as a smaller sample size, intraspecific tick characteristics, or low-level pathogen infection in the analysed ticks, which may hinder pathogen detection and lead to an underestimation of pathogen prevalence [57].

The presence of *R. aeschlimannii* and *C. burnetii* in *H. lusitanicum* questing ticks suggests that these pathogens may also be present in wild rabbits and ungulates. This possibility is further supported by the detection of *C. burnetii* in other Mediterranean populations [16]. The wide diversity of potential hosts and the close contact between livestock and humans in the Maamora forest, which may become infested by questing ticks, increases pathogen transmission opportunities and the possible infection of wild and domestic animals [50,58,59]. This also poses a risk to humans [48], favouring zoonotic infections (re)emergence in North Africa [20].

Despite advances in the study of tick-host-pathogen interactions in Northern Africa, this study presents some limitations to be considered. The timeframe of the study, a little more than a year, may limit the ability to depict and fully comprehend the tick-host dynamics. In addition, due to budget constraints, we were unable to analyse tick-borne pathogens in ticks infesting rabbits, which could provide additional information into disease transmission dynamics.

To understand how future climate scenarios may impact the distribution and ecology of ticks and tick-borne diseases in Northern Africa, prolonged fieldwork is required to elucidate the complex nature of ticks, including questing-wild hosts-pathogen interactions. Enhanced surveillance is necessary to detect and control North African ticks originating from the importation of animals (livestock) and exotic pet trade (e.g., tortoises) to prevent the inadvertent establishment of exotic North African tick species in Europe. In this line, the movement of ticks between Europe and North Africa presents a challenge, not only to human health, but also to animals and ecosystems [20]. Conservation and management programs could bring together researchers and policymakers from both European and North African countries, fostering collaboration through a One Health approach.

## Supporting information

**S1 Fig. Seasonal rabbit density (individual/km$^2$) in the four different zones in the period 2022–2023.**
(TIF)

**S2 Fig. Seasonal aggregation areas of rabbits in the period 2022–2023.**
(TIF)

**S1 Table. List of analysed samples, DNA/RNA extraction results, as well as PCR infectious agents detected.**
(XLSX)

**S2 Table. List of models organized by their Akaike Information Criteria (AIC).**
(DOCX)

## Acknowledgments

We thank Hassan Belhajjamia, Abdessalam Belhajjamia and Bouhali Kaddouri for their field assistance in the collection of ticks. We would also like to thank Greg Trollip and Jacob Mwanzia for their support and interest in wild species conservation. Finally, we would like to sincerely thank Alfonso Peralbo-Moreno for his invaluable assistance in the identification of ticks, which greatly contributed to the success of this study.

## Author contributions

**Conceptualization:** Oscar Rodríguez, Christian Gortázar.

**Data curation:** Marta Rafael, Amalia Segura, Oscar Rodríguez.

**Formal analysis:** Marta Rafael, Amalia Segura.

**Investigation:** Marta Rafael, David Relimpio.

**Methodology:** Marta Rafael, Rita Vaz-Rodrigues.

**Supervision:** José de la Fuente.

**Writing – original draft:** Marta Rafael, Amalia Segura.

**Writing – review & editing:** Rita Vaz-Rodrigues, Oscar Rodríguez, Gabriela de la Fuente, Julio Isla, Christian Gortázar, José de la Fuente.

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
