## [Decision Letter · Decision Letter 0]

Dear Dr. Rafael,

Write Species names in the phylogenetic tree in italics "It's a must"

These are some edits also

'Corresponding Authors' is written 'Autors'

* Introduction

- The introduction provides sufficient background and well-written study objectives. However, it would be better to add how to investigate with technical methods in the introduction.

- Line 74: Hyalomma (H.) aegyptium

- Line 109: This sentence is quite objective.

- Line 96: Why did you choose only wild rabbits?

* Materials and Methods

- The materials and methods presented details of the overall studies.

- However, it would be better to abbreviate the study site and the tick sampling part.

- Please describe how ticks collected from animals were handled.

* Results

- Line 234: Why did you indicate as "spp."?

* Discussion

- The discussion is qualified in terms of evaluating the results. However, the discussion is limited to evaluating the results. It would be better to compare with similar studies conducted worldwide using the same or similar techniques, which would enrich the text. It is recommended to include an evaluation of the results of recent similar studies conducted around the world in the discussion.

We look forward to receiving your revised manuscript.

Kind regards,

Dina Aboelsoued, Ph.D.

Academic Editor

PLOS ONE

Journal Requirements:

https://www.sciencedirect.com/science/article/pii/S2352771423000678?via%3Dihub

https://pubmed.ncbi.nlm.nih.gov/37973690/

https://link.springer.com/article/10.1007/s10493-023-00863-7

In your revision ensure you cite all your sources (including your own works), and quote or rephrase any duplicated text outside the methods section. Further consideration is dependent on these concerns being addressed.

4. Thank you for stating in your Funding Statement: [This research was partly supported by the 2022-GRIN-34227 grant, funded by the University of Castile-La Mancha (UCLM), Spain and EU-FEDER. R. Vaz-Rodrigues was supported by a doctoral contract (2022-PRED-20675), from UCLM and co-financed by the European Social Fund (ESF).].

5. Please include captions for your Supporting Information files at the end of your manuscript, and update any in-text citations to match accordingly. Please see our Supporting Information guidelines for more information: http://journals.plos.org/plosone/s/supporting-information .

Additional Editor Comments:

* Introduction

- The introduction provides sufficient background and well-written study objectives. However, it would be better to add how to investigate with technical methods in the introduction.

- Line 74: Hyalomma (H.) aegyptium

- Line 109: This sentence is quite objective.

- Line 96: Why did you choose only wild rabbits?

* Materials and Methods

- The materials and methods presented details of the overall studies.

- However, it would be better to abbreviate the study site and the tick sampling part.

- Please describe how ticks collected from animals were handled.

* Results

- Line 234: Why did you indicate as "spp."?

* Discussion

- The discussion is qualified in terms of evaluating the results. However, the discussion is limited to evaluating the results. It would be better to compare with similar studies conducted worldwide using the same or similar techniques, which would enrich the text. It is recommended to include an evaluation of the results of recent similar studies conducted around the world in the discussion.

Reviewers' comments:

Reviewer's Responses to Questions

**Comments to the Author**

1. Is the manuscript technically sound, and do the data support the conclusions?

Reviewer #1: Yes

Reviewer #2: Yes

Reviewer #3: Yes

2. Has the statistical analysis been performed appropriately and rigorously?

Reviewer #1: I Don't Know

Reviewer #2: Yes

Reviewer #3: Yes

3. Have the authors made all data underlying the findings in their manuscript fully available?

Reviewer #1: Yes

Reviewer #2: Yes

Reviewer #3: Yes

4. Is the manuscript presented in an intelligible fashion and written in standard English?

Reviewer #1: Yes

Reviewer #2: Yes

Reviewer #3: Yes

Reviewer #1: * Introduction

- The introduction provides sufficient background and well-written study objectives. However, it would be better to add how to investigate with technical methods in the introduction.

- Line 74: Hyalomma (H.) aegyptium

- Line 109: This sentence is quite objective.

- Line 96: Why did you choose only wild rabbits?

* Materials and Methods

- The materials and methods presented details of the overall studies.

- However, it would be better to abbreviate the study site and the tick sampling part.

- Please describe how ticks collected from animals were handled.

* Results

- Line 234: Why did you indicate as "spp."?

* Discussion

- The discussion is qualified in terms of evaluating the results. However, the discussion is limited to evaluating the results. It would be better to compare with similar studies conducted worldwide using the same or similar techniques, which would enrich the text. It is recommended to include an evaluation of the results of recent similar studies conducted around the world in the discussion.

Reviewer #2: Ticks and tick-borne pathogens pose a serious threat worldwide. The study offers important findings that will help protect both human and wildlife health. I have some edits and comments in the attached pdf file for the authors to consider.

Reviewer #3: The study focuses on a timely and highly relevant topic: the biodiversity of ticks in a natural Mediterranean area, their seasonal ecology, the interactions between ticks and wild hosts, and the presence of important zoonotic pathogens. The significance of the research is high, considering the role of ticks in disease transmission and the potential impact of climate change on the dynamics of arthropod vector populations.

The work is generally well-structured: the introduction, methods, and results are adequately described. The discussion effectively contextualizes the findings within the ecological and health framework of the region, providing a valuable overview of the importance of seasonal and environmental dynamics in the distribution of ticks and pathogens.

References to the existing literature are appropriate and contribute to reinforcing hypotheses regarding host-vector interactions. The focus on climate change and its possible consequences on tick distribution is a strong point, aligning with the growing concern about emerging zoonotic risks.

The study’s limitations are acknowledged, particularly regarding the limited sampling timeframe and the lack of pathogen analyses in rabbits.

The emphasis on a One Health approach is rightly highlighted as necessary to address the complexity of the issue.

**Do you want your identity to be public for this peer review?** For information about this choice, including consent withdrawal, please see our Privacy Policy

Reviewer #1: No

Reviewer #2: No

Reviewer #3: No

---

## [Author Response · Author response to Decision Letter 1]

10 Jun 2025

Reviewer #1:

* Introduction

- The introduction provides sufficient background and well-written study objectives. However, it would be better to add how to investigate with technical methods in the introduction.

Methods to achieve the proposed objectives have been added to the introduction as: “i) determine the abundance and diversity of questing ticks by flagging and ticks collected from dead rabbits seasonally, ii) determine density-dependent and environmental effects in questing ticks and ticks from wild rabbits, and iii) determine tick-borne pathogens circulating in questing ticks and those infecting addax. Tick-borne pathogens were detected by conventional PCR techniques, aimed at Anaplasma spp., Babesia spp., C. burnetii, Rickettsia spp., Theileria spp. and CCHFV, previously described pathogens in Africa [2,11,19,20] and in the study area [11,14], followed by a phylogenetic analysis of positive results.” (see lines 108-115 in manuscript with track changes).

- Line 74: Hyalomma (H.) aegyptium

Thank you very much, it has been corrected.

- Line 109: This sentence is quite objective.

The sentence has been rewritten: “This study results provide information to advance in a One-Health approach aimed at preventing the transmission of zoonotic diseases across trans-Mediterranean regions impacting both Europe and Northern Africa [20].”

- Line 96: Why did you choose only wild rabbits?

Thank you very much for this question. We were expecting the presence of Hyalomma lusitanicum and H. marginatum ticks, which had been previously detected in the study area. Wild rabbits were selected due to their influence in the tick’s life cycle, for being highly abundant and for its long-term monitoring program in the area. In the future, we are planning to study more species such as the Barbary deer and evaluate the role of other possible hosts and their effect on the prevalence of tick-borne pathogens.

* Materials and Methods

- The materials and methods presented details of the overall studies.

- However, it would be better to abbreviate the study site and the tick sampling part.

Thank you for this comment. The study site information (line 123) and tick sampling (line 160) have been summarized.

- Please describe how ticks collected from animals were handled.

Collected ticks from vegetation and animals were handled in the same manner. As not to repeat the same information three times in the Tick sampling description, a final sentence was added (line 186-188) describing the method employed for all collected ticks:

“Ticks collected from vegetation, wild rabbits and addax were placed into plastic vials containing RNAlater (Thermo Fisher, Waltham, MA, USA) and stored at -20 ºC until further analysis.”

* Results

- Line 234: Why did you indicate as "spp."?

When doing morphological identification of questing ticks using a stereomicroscope, we were not able to determine with certainty the Rhipicephalus tick’s species. Nevertheless, to solve this issue, the tick's DNA was extracted, and its molecular identification was performed, allowing its classification as Rhipicephalus pusillus (mentioned in line 321).

* Discussion

- The discussion is qualified in terms of evaluating the results. However, the discussion is limited to evaluating the results. It would be better to compare with similar studies conducted worldwide using the same or similar techniques, which would enrich the text. It is recommended to include an evaluation of the results of recent similar studies conducted around the world in the discussion.

Thank you for this comment, in the discussion it has been added additional information discussing tick-borne rickettsiosis and Coxiella burnetii in Africa and Mediterranean areas. The following sentences have been added to the manuscript:

“Studies conducted in other regions of North Africa point to a prevalence of tick-borne rickettsiosis between 0.5 - 64%, depending on the Rickettsia species [20]” (lines 449-451)

“We detected a lower prevalence of C. burnetii (22.6% of questing ticks and 28.6% of addax ticks), compared to previous studies, which describe higher prevalences in the Mediterranean basin [16,53] or African countries [20]. Such differences may be attributed to different tick characteristics, the multitude of tick species and host communities present in both regions, which could condition the maintenance and/or increase the prevalence of the pathogen [1].” (lines 458-464).

Reviewer #2:

Ticks and tick-borne pathogens pose a serious threat worldwide. The study offers important findings that will help protect both human and wildlife health. I have some edits and comments in the attached pdf file for the authors to consider.

Thank you very much for your comments. Edits and suggestions have all been accepted. Comments have been addressed in the following matter.

- Line 187: Collected from Addax?

Sorry for not having been clear in the initial manuscript. Due to budget constrains only ticks from vegetation and rabbit samples were sequenced. Morphologically, most collected ticks from addax and vegetation were identified as Hyalomma lusitanicum, hence we only confirm the identification with vegetation ticks. The sentence was changed for: “Ticks used for molecular identification included four individual adult ticks collected from vegetation, confirming also the identification of addax’s ticks, and ticks collected from rabbits - four individual adult ticks, and three pools of nymphs, constituted by two to four nymphs.” (See lines 207-210)

- Line 236: Looks like the one nymph was excluded. Confirm and change this to 1230

Thank you for noticing this mistake. The number of ticks has been updated to 1230.

-Line 239: The numbers add up to 3753. It was previously stated that questing ticks were 3754. Check and revise.

Thank you once again for pointing out the error. There was a tick missing in zone D. The sentence has been modified to: “Spatially, more questing ticks were collected in zones C and A (1948 and 1127, respectively) than in B and D (631 and 48, respectively).” (see lines 263-264).

- Line 288: You used the molecular method to confirm the species of the adult and nymph ticks. However, the larvae were not mentioned. Did you use the molecular method to confirm the tick species that were larvae? Larvae are usually difficult to identify using morphology.

Molecular identification was only performed in adult and nymph species. There was a mistake in the classification of larvae ticks collected from rabbits, as larvae ticks were only identified up to the genus level. This issue has been corrected as: “Ticks were morphologically classified as H. lusitanicum (4 adults), Hyalomma spp. (347 nymphs and larvae) and Rhipicephalus pusillus (583 adults, 185 nymphs), and Rhipicephalus spp. (18 larvae).” (see line 280).

- Line 329: This sentence is wordy. Can it be rephrased?

Thank you for this comment. The sentence was rewritten to: “The 16S rDNA PCR-positive sample (Isolate 23) to Rickettsia spp. shared 100% nucleotide identity with R. aeschlimannii retrieved in Spain from H. rufipes and H. marginatum ticks (GenBank accession numbers MW394393 and MW394392, respectively) (Fig 6).” (see lines 355-359).

- Line 407: Was a thorough search done to confirm this is the first worldwide report? If not, its best you limit it to North Africa or Africa.

Thank you for this comment. A thorough search was not performed, and although no articles were found on Rickettsia aeschlimannii in Hyalomma lusitanicum questing ticks during the bibliographic search, we are not able to confirm this is the first detection worldwide. Hence, the sentence has been changed to: “To our knowledge, we report the first detection of R. aeschlimannii in H. lusitanicum questing ticks in Africa, along with the identification of C. burnetii.” (see lines 442-444).

Reviewer #3:

The study focuses on a timely and highly relevant topic: the biodiversity of ticks in a natural Mediterranean area, their seasonal ecology, the interactions between ticks and wild hosts, and the presence of important zoonotic pathogens. The significance of the research is high, considering the role of ticks in disease transmission and the potential impact of climate change on the dynamics of arthropod vector populations. The work is generally well-structured: the introduction, methods, and results are adequately described. The discussion effectively contextualizes the findings within the ecological and health framework of the region, providing a valuable overview of the importance of seasonal and environmental dynamics in the distribution of ticks and pathogens.

References to the existing literature are appropriate and contribute to reinforcing hypotheses regarding host-vector interactions. The focus on climate change and its possible consequences on tick distribution is a strong point, aligning with the growing concern about emerging zoonotic risks. The study’s limitations are acknowledged, particularly regarding the limited sampling timeframe and the lack of pathogen analyses in rabbits. The emphasis on a One Health approach is rightly highlighted as necessary to address the complexity of the issue.

Thank you very much for your comment. It is highly appreciated.

Response to the editors’ concerns:

Thank you very much for the comments and improvements on the manuscript. All comments have been addressed, and a detail description is presented on each comment. An additional sentence has been included in the acknowledgements, that we would appreciate if it would be included in the final version (lines 506-510).

Manuscript needs language editing as some words are spelled wrong and some phrases lack proper English language construction "It's a must"

The manuscript has been revised to correct its English and misspelled words.

Write Species names in the phylogenetic tree in italics "It's a must"

Thank you for this comment, this has been emended.

Figures are not clear and need to be resubmitted after resolution adjustment "It's a must"

We agree, figures have been corrected according to specified comments. Species names have been corrected to italic and figures resolution has been improved.

The whole manuscript should be shortened especially the materials and Methods section which contains many Introduction sentences

According to reviewers' comments, we have shorted the manuscript.

These are some edits also

'Corresponding Authors' is written 'Autors'

Misspelled words have been addressed.

Manuscript and file name style have been modified according to PLOS ONE requirements.

It has been included a paragraph in the methodology addressing this comment in lines 189-192 as: “Ticks’ collection from vegetation, as well as from dead and live animals, was carried out as part of a private initiative for monitoring threatened species in the Maamora Forest, following approved ethical protocols for wildlife capture and management (no permits were required).” The protocols employed followed international standards.

https://www.sciencedirect.com/science/article/pii/S2352771423000678?via%3Dihub

https://pubmed.ncbi.nlm.nih.gov/37973690/

https://link.springer.com/article/10.1007/s10493-023-00863-7

In your revision ensure you cite all your sources (including your own works), and quote or rephrase any duplicated text outside the methods section. Further consideration is dependent on these concerns being addressed.

Thank you for this comment, this issue has been addressed by rephasing sentences and including the reference of mentioned articles.

4. Thank you for stating in your Funding Statement: [This research was partly supported by the 2022-GRIN-34227 grant, funded by the University of Castile-La Mancha (UCLM), Spain and EU-FEDER. R. Vaz-Rodrigues was supported by a doctoral contract (2022-PRED-20675), from UCLM and co-financed by the European Social Fund (ESF).].

An amendment of the Funding Statement has been included in the manuscript as, “This research was supported by the 2022-GRIN-34227 grant, funded by the University of Castile-La Mancha (UCLM), Spain and EU-FEDER. R. Vaz-Rodrigues was supported by a doctoral contract (2022-PRED-20675), from UCLM and co-financed by the European Social Fund (ESF). There was no additional external funding received for this study.”

Captions of supporting material have been included after the references.

References have been actualized according to retracted or additional text, following the reviewers’ comments. Previous articles cited with the numbers 25 and 26 have been retracted. Eliminated references include the articles:

25. Segura A, Rotger A, Rodríguez-Caro RC. Hidden Threats to Persistence: Changes in Population Structure Can Affect Well-Preserved Spur-Thighed Tortoise Populations. Herpetologica. 2025;81: 23–33. doi:10.1655/Herpetologica-D-23-00066

26. Filahi S, Tanarhte M, Mouhir L, El Morhit M, Tramblay Y. Trends in indices of daily temperature and precipitations extremes in Morocco. Theor Appl Climatol. 2016;124: 959–972. doi:10.1007/s00704-015-1472-4

In addition, in the revised manuscript, reference 53 has been added:

53. González J, González MG, Valcárcel F, Sánchez M, Martín-Hernández R, Tercero JM, et al. Prevalence of Coxiella burnetii (Legionellales: Coxiellaceae) Infection Among Wildlife Species and the Tick Hyalomma lusitanicum (Acari: Ixodidae) in a Meso-Mediterranean Ecosystem. J Med Entomol. 2020;57: 551–556. doi:10.1093/jme/tjz169

The final list of refences is as follows:

1. de la Fuente J, Contreras M, Estrada-Peña A, Cabezas-Cruz A. Targeting a global health problem: Vaccine design and challenges for the control of tick-borne diseases. Vaccine. 2017;35: 5089–5094. doi:10.1016/j.vaccine.2017.07.097

2. Estrada-Peña A, Mihalca AD, Petney TN. Ticks of Europe and North Africa: A Guide to Species Identification. Springer; 2017.

3. Sonenshine DE, Roe RM. Biology of Ticks Volume 1. OUP USA; 2013.

4. Allan BF, Dutra HP, Goessling LS, Barnett K, Chase JM, Marquis RJ, et al. Invasive honeysuckle eradication reduces tick-borne disease risk by altering host dynamics. Proc Natl Acad Sci U S

---

## [Decision Letter · Decision Letter 1]

Tick-wildlife host-pathogen network interactions in Northern Africa

PONE-D-25-17487R1

Dear Ms Marta Simoes Correia Rafael,

We’re pleased to inform you that your manuscript has been judged scientifically suitable for publication and will be formally accepted for publication once it meets all outstanding technical requirements.

Kind regards,

Dina Aboelsoued, Ph.D.

Academic Editor

PLOS ONE

Reviewers' comments:

Reviewer's Responses to Questions

**Comments to the Author**

Reviewer #2: All comments have been addressed

2. Is the manuscript technically sound, and do the data support the conclusions?

Reviewer #2: Yes

3. Has the statistical analysis been performed appropriately and rigorously?

Reviewer #2: Yes

4. Have the authors made all data underlying the findings in their manuscript fully available?

Reviewer #2: Yes

5. Is the manuscript presented in an intelligible fashion and written in standard English?

Reviewer #2: Yes

Reviewer #2: (No Response)

**Do you want your identity to be public for this peer review?** For information about this choice, including consent withdrawal, please see our Privacy Policy

Reviewer #2: No

---

## [Editor Report · Acceptance letter]

PONE-D-25-17487R1

PLOS ONE

Dear Dr. Rafael,

I'm pleased to inform you that your manuscript has been deemed suitable for publication in PLOS ONE. Congratulations! Your manuscript is now being handed over to our production team.

Kind regards,

on behalf of

Dr. Dina Aboelsoued

Academic Editor

PLOS ONE